# Descriptive Study on the Relationship between Dyspnea, Physical Performance, and Functionality in Oncology Patients

**DOI:** 10.3390/healthcare12161675

**Published:** 2024-08-22

**Authors:** Diego Lucas-Ruano, Celia Sanchez-Gomez, María Isabel Rihuete-Galve, Alberto Garcia-Martin, Emilio Fonseca-Sanchez, Eduardo José Fernández-Rodríguez

**Affiliations:** 1Department of Nursing and Physiotherapy, Universidad de Salamanca, 37008 Salamanca, Spain; diegolucsru@usal.es (D.L.-R.); rihuete@usal.es (M.I.R.-G.); 2Institute of Biomedical Research of Salamanca (IBSAL), 37007 Salamanca, Spain; celiasng@usal.es; 3Department of Developmental and Educational Psychology, Institute of Biomedical Research of Salamanca (IBSAL), University of Salamanca, 37008 Salamanca, Spain; efonseca@usal.es; 4Department of Medical Oncology, University Hospital of Salamanca, Institute of Biomedical Research of Salamanca (IBSAL), 37007 Salamanca, Spain; albergm@usal.es; 5Department of Labour Law and Social Work, Institute of Biomedical Research of Salamanca (IBSAL), University of Salamanca, 37007 Salamanca, Spain

**Keywords:** cancer, nursing, care, dyspnea, physical performance

## Abstract

Background: Cancer is a leading cause of morbidity and mortality globally. Dyspnea, affecting up to 60% of cancer patients, exacerbates physical and psychological distress, reducing quality of life. This study aims to explore the relationship between dyspnea and factors such as age, sex, clinical diagnosis, and treatment lines in cancer patients, with the goal of improving understanding and management of this debilitating symptom to enhance patient care and quality of life. Methods: This study employed an observational, cross-sectional, and descriptive approach to investigate patients with oncological disease at the University Hospital of Salamanca between March 2021 and April 2024. A convenience sample was selected, including patients over 18 years old with a pathological diagnosis of cancer, experiencing any degree of dyspnea, and who consented to participate by signing the informed consent. Exclusion criteria included lack of consent and clinical conditions that prevented an interview. The studied variables encompass sociodemographic (age, gender, diagnosis, tumor stage, number of treatment lines) and clinical aspects (daily activities, degree of dyspnea, functional capacity, physical performance), evaluated using the Barthel Index, the mMRC Dyspnea Scale, the ECOG Scale, and the Short Physical Performance Battery (SPPB). Data were collected through semistructured interviews and medical records, and analyzed using specialized software. This research has ethical approval CEiM Code 2023 12 1472, Reference 2024/01. Results: The mean age was 66.82 years. Lung cancer was predominant (60.2%), with most patients in stage 3 (65.7%) and receiving three treatment lines (68.7%). Higher age, advanced disease stage, and more treatment lines correlated with lower Barthel and SPPB scores, and higher ECOG and mMRC scores, indicating worse functionality, physical performance, and greater dyspnea. No significant correlations were found between gender or pathological diagnosis and the studied variables. Conclusions: Advanced age, higher disease stage, and more treatment lines are associated with decreased functionality, poorer physical performance, and increased dyspnea in cancer patients. Gender and specific cancer diagnosis do not significantly affect these relationships. Addressing dyspnea is crucial to improving the quality of life and physical performance in this population. Future studies should explore additional factors like treatment types and nutritional status.

## 1. Introduction

Cancer is a disease in which some of the body’s cells multiply uncontrollably, spread to other tissues and organs, and form cell masses called tumors [1].

Globally, cancer remains one of the leading causes of morbidity and mortality. According to estimates by the International Agency for Research on Cancer (IARC), approximately 18.1 million new cancer cases were diagnosed in 2020, and this figure is expected to rise to 28.0 million in the coming decades [2]. In Spain, it is estimated that there will be 286,664 new cancer cases, with colon and rectal, breast, lung, prostate, and urinary bladder cancers being the most common. In 2022, the most diagnosed cancers were breast, lung, and colon and rectal cancers [3].

Cancer survival refers to the probability of surviving after a certain time from diagnosis, excluding other causes of death. In Spain, between 2008 and 2013, the five-year survival rate was 55.3% in men and 61.7% in women, differences that can be attributed to the prevalence of certain tumors in each gender [2].

Despite the reduction in mortality rates and improvements in treatments, many cancer patients suffer from side effects and sequelae. Among the most common are decreased functional and pulmonary capacity, fatigue, nausea, vomiting, pain, dyspnea, insomnia, loss of appetite, constipation, diarrhea, drowsiness, hair loss, mouth pain, and sweating [4].

Dyspnea, defined by the American Thoracic Society (ATS) as a subjective experience of breathing discomfort, affects up to 60% of cancer patients. It can manifest as resting dyspnea, exertional dyspnea, paroxysmal dyspnea, orthopnea (disappears when standing or sitting), and platypnea (increases when standing or sitting). The pathophysiology of dyspnea is complex and may involve cerebropulmonary mechanisms that, when desynchronized, generate symptoms such as shortness of breath, chest pressure, and increased breathing effort [5].

Various studies have identified risk factors for dyspnea, such as advanced age, female sex, high body mass index, and heart or respiratory diseases. In the context of cancer, multiple factors can induce dyspnea, including lung cancer, malignant pleural effusion, and liver enlargement or ascites, which elevate the diaphragm and decrease lung volume [6,7]. Furthermore, certain cancer treatments can also cause dyspnea. Thoracic surgery, common in lung cancer patients, can increase respiratory load by altering pulmonary surfactant characteristics. Thoracic radiotherapy can progressively damage the lungs, worsening gas exchange capacity. Drug-induced lung diseases, such as pneumonitis or pulmonary fibrosis, also increase dyspnea by reducing arterial oxygen pressure. Anemia, present in 30–90% of cancer patients, decreases hemoglobin and oxygen levels in the blood, activating chemoreceptors that increase ventilation and breathing effort [8].

Dyspnea, besides its physical component, is closely related to psychological stress. It causes significant discomfort, reducing quality of life and associating with anxiety and depression. Unpredictable dyspnea episodes are particularly distressing and stressful, causing feelings of loss of control and fear.

Dyspnea severely affects patients’ quality of life, especially those in palliative care, where it is associated with poorer quality of life and higher symptom burden. Quality of life (QoL), according to Modesto Silveira and collaborators, is defined as “the individual’s perception of their position in life, within the cultural system and values in which they live and in relation to their goals, expectations, norms, and concerns” [9].

As the number of cancer survivors increases, quality of life and functionality often deteriorate due to treatment side effects. Struggling with persistent symptoms can negatively impact quality of life, physical, social, and cognitive functioning, and increase healthcare costs.

Maximizing occupation and quality of life during the survival period is crucial. A study by Pergolotti and collaborators indicated that many cancer survivors consider functional outcomes as important or more important than overall survival. Dyspnea is especially prevalent in lung cancer patients, with 75% developing it and 90% experiencing it one month before death [10].

Patients with colon and rectal cancer can also experience dyspnea, especially if they have cardiovascular diseases. Functional status is vital, especially in elderly cancer patients, and relates to survival and quality of life. The decline in general physical function and health-related quality of life (HRQoL) is often attributed to low exercise capacity, leading to a sedentary lifestyle and increased risk of complications.

Facing deficiencies and functional limitations can be a direct consequence of cancer or toxic treatments. According to a study by Weaver and collaborators, cancer survivors reported worse physical and emotional quality of life compared to the general population. Decreased exercise capacity affects tolerance to treatments such as chemotherapy and radiotherapy, compromising survival [11].

Recent research, such as that by Fernández and collaborators, highlights how dyspnea negatively impacts the quality of life and daily activities of patients with advanced cancer. This symptom, especially in lung cancer patients, is a crucial indicator of the need for timely interventions to improve quality of life [12,13].

This study arises from the need to explore and understand the relationship between the risk of dyspnea and its impact on the functionality and physical performance of cancer patients, factors that culminate in a deterioration of quality of life. This need is based on the limited literature available on the subject. Various studies have indicated that dyspnea creates a harmful cycle: the improvement in the effectiveness of cancer treatments increases overall survival, but also increases the prevalence of dyspnea. This situation leads to increased exertional dyspnea and reduced physical exercise, resulting in lower treatment tolerance and decreased survival, which in turn increases healthcare demand.

Based on previous studies, our initial hypothesis proposes that there is a significant correlation between various parameters, such as age, clinical diagnosis, number of treatment lines, and the patient’s general condition, with the presence of dyspnea, physical performance, and functionality in cancer patients. This study aims to clarify these links to contribute to a better understanding and management of dyspnea in the oncological context, thereby improving patients’ quality of life and comprehensive care.

This study aims to investigate the relationship between various parameters and dyspnea, physical performance, and functionality in cancer patients. Specifically, it seeks to determine if there is a relationship between the degree of dyspnea and the deterioration of physical performance, evaluate the relationship between the presence of dyspnea and the patient’s age, examine the possible correlation between the patient’s sex and the occurrence of dyspnea, analyze the influence of the pathological diagnosis on the incidence of dyspnea, and investigate whether the number of treatment lines affects the prevalence of dyspnea in cancer patients.

## 2. Materials and Methods

### 2.1. Design and Procedures

An observational, descriptive, cross-sectional study was designed to meet the objectives.

### 2.2. Participants

Patients were selected through a convenience sample, with invitations extended to those receiving active treatment in the Oncology Ward and Day Hospital at the University Hospital of Salamanca to participate. The research was conducted between March 2021 and April 2024.

The sample size was estimated based on similar studies by Muruganandan et al. [13]; in the study (n = 170), the management of breakthrough dyspnea in cancer patients was investigated, including a sample size of approximately 170 patients. This study aimed to describe the characteristics of cancer patients experiencing breakthrough dyspnea and the attributes of the disorder. Conducted between May 2015 and March 2016, the study provided insights into the prevalence and management practices for dyspnea in this patient population, highlighting the significant impact of dyspnea on the quality of life and the need for effective management strategies.

The following criteria were used to select the sample:Inclusion Criteria
−Patients admitted to the Oncology Ward and Day Hospital at the University Hospital of Salamanca.−Anatomopathological diagnosis of oncological disease.−Patients presenting any degree of dyspnea.−Patients over 18 years old.−Patients who agree to participate by signing the informed consent.−Patients who meet these criteria and wish to participate, after understanding and signing the informed consent, will be included in the study.
Exclusion Criteria
−Patients whose clinical condition does not allow for an interview.
Study Variables

Sociodemographic variables: Age, gender, anatomopathological diagnosis, tumor stage, number of treatment lines.

Study Variables

−Performance of activities of daily living (ADL).−Degree of dyspnea.−Functional capacity.−Physical performance.

Assessment Instruments

To evaluate the variables of the study, the following questionnaires were used:

Barthel Index (BI): Assesses physical performance in ADLs through a 10-item survey covering toilet use, bathing, dressing, grooming, bladder and bowel control, climbing, walking, chair transfer, and feeding. Scores range from 0 (completely dependent) to 100 (completely independent) [14].

Modified Medical Research Council (mMRC) Dyspnea Scale: Classifies patients into 5 grades, from 1 (normal state) to 5 (severe dyspnea). The ATS modification focuses on dyspnea produced during walking, evaluating specific activities [15].

Eastern Cooperative Oncology Group (ECOG) Scale: Assesses the functionality of cancer patients in five levels, from 0 (normal functioning) to 5 (death) [16].

Short Physical Performance Battery (SPPB): Evaluates the functional performance of lower limbs through timed tests of walking speed, standing balance, and lower limb strength. Scores range from 0 to 12, from worst to best performance [17].

The data for the independent variables were obtained from the patient’s medical record.

Procedure

The study commenced once the required number of participants was reached. Patients provided informed consent, and semistructured, informal interviews were conducted either in the inpatient unit rooms or a room in the Oncology Day Hospital. These interviews, conducted by experienced professionals from the Medical Oncology Service and the Institute of Biomedical Research (IBSAL) in Salamanca, Spain, aimed to prevent tedium and maintain engagement. Personal data and details about the oncological process and treatments received, as well as evaluation scales, were collected. This information was recorded using specialized software and analyzed later, utilizing a database created in Microsoft Access 2021.

Statistical Methodology

Descriptive Methodology

For quantitative variables with a normal distribution, the mean and standard deviation were used. For variables that did not follow a normal distribution, the median and quartiles were employed. The distribution of the variables was confirmed using the Kolmogorov–Smirnov test, and data were considered to follow a normal distribution if the *p*-value was greater than 0.05.

Analytical Methodology

To analyze two quantitative variables, Pearson’s correlation coefficient was used. To evaluate two qualitative variables, the chi-square statistical test was employed. To analyze the relationship between a qualitative variable and a quantitative variable, the Student’s *t*-test was used if the sample size was *n* < 30, and a normal statistical test was used if the sample size was *n* > 30. The ANOVA statistical test was used to analyze the relationship between a qualitative variable (with more than two categories) and a quantitative variable.

Data Processing

Data analysis was performed using the statistical program SPSS, version 28.

## 3. Results

### 3.1. Sample and Demographic Characteristics

A total of 166 patients meeting the inclusion criteria were included in the study. The mean age was 66.82 years (SD = 10.420), ranging from 50 to 91 years. The mean age for males was 66.62 years (n = 85, SD = 10.111, range 51–85), and for females, it was 67.02 years (n = 81, SD = 10.793, range 50–91). Males comprised 51.2% of the sample, while females accounted for 48.8%.

### 3.2. Diagnosis and Treatment

The predominant types of tumors were lung (60.2%), digestive system (19.9%), breast (15.1%), central nervous system and other tumors (1.8% each), and prostate (1.2%). Most patients received three lines of treatment (68.7%), followed by four lines (23.5%), two lines (4.8%), five lines (1.8%), and six lines (1.2%).

### 3.3. Disease Stage

Most patients were in stage 3 (65.7%), followed by stage 4 (33.1%) and stage 2 (1.2%).

### 3.4. Analysis of Study Variables

After the Kolmogorov–Smirnov test confirmed that the data followed a normal distribution (with a *p*-value greater than 0.05), the mean of each variable under study was calculated. The results showed that the average score of the Barthel questionnaires was 34.19, with a standard deviation of 12.089, a minimum of 10, and a maximum of 55. For the ECOG questionnaires, the mean was 2.83, with a standard deviation of 0.666, a minimum of 2, and a maximum of 4. Meanwhile, the average score of the MRC questionnaires was 2.80, with a standard deviation of 0.715, a minimum of 1, and a maximum of 4. Lastly, the average score of the SPPB questionnaires was 5.66, with a standard deviation of 2.041.

To analyze the relationship between two quantitative variables, Pearson’s correlation coefficient was used. The following associations were identified: age showed an inverse correlation with the Barthel score (r = −0.543), suggesting that older age is associated with lower Barthel scores. On the other hand, age showed a direct correlation with the ECOG (r = 0.561) and MRC (r = 0.574) scores, indicating that older age is associated with higher ECOG scores and increased levels of dyspnea according to MRC. Additionally, an inverse correlation was found between age and the SPPB score (r = −0.594), suggesting that older individuals tend to have lower SPPB scores.

Furthermore, the relationship between the stage of the disease and the different variables was explored. An inverse correlation was found between the disease stage and the Barthel score (r = −0.373), indicating that as the disease progresses, Barthel scores decrease. A direct correlation was also observed between the disease stage and the ECOG (r = 0.276) and MRC (r = 0.267) scores, suggesting that as the disease advances, ECOG scores and dyspnea levels according to MRC tend to increase. Lastly, an inverse correlation was identified between the disease stage and the SPPB score (r = −0.282), suggesting that more advanced stages of the disease are associated with lower SPPB scores.

Finally, the relationship between the number of treatment lines and the variables under study was examined. It was observed that a higher number of treatment lines is associated with lower Barthel (r = −0.266) and SPPB (r = −0.137) scores, and with higher ECOG (r = 0.220) and MRC (r = 0.222) scores, indicating that a greater number of treatment lines is related to poorer functional status and increased dyspnea.

### 3.5. Analysis of Qualitative Variables

For the analysis of a qualitative variable with two categories and a quantitative variable, the T-TEST was employed. The analysis did not reveal any statistically significant differences related to gender, indicating no established relationship.

For the analysis of a qualitative variable with more than two categories and a quantitative variable, the ANOVA test was used. The analysis did not show any statistically significant differences concerning the anatomopathological diagnosis compared to other variables, indicating no established relationship (Table 1, Table 2, Table 3 and Table 4).

## 4. Discussion

In this study, we aimed to observe the relationship between various parameters and dyspnea, physical performance, and functionality in cancer patients. The Barthel Index and Short Physical Performance Battery (SPPB) indicate that functionality and physical performance scores are lower in older patients, with more advanced stages and a higher number of treatment lines. Comparing our results with prospective studies such as Leach et al. (2023), which explores the relationship between pathological diagnosis and the ability to perform activities of daily living (ADLs), it was observed that aging combined with cancer affects a greater variety and number of functions, which is consistent with our findings [18].

However, our study did not find a significant relationship between pathological diagnosis and the level of functionality or physical performance. This contrasts with other studies indicating that, in lung cancer patients, activities such as bathing, dressing, feeding, and standing are affected, and that this impairment is also seen in breast and colorectal cancer. The literature suggests that age is a critical factor for evaluating the use of the SPPB, given that the older population is highly vulnerable and sedentary behavior increases the risk of disability [19].

Our study revealed a clear association between advanced age and a slight decrease in physical performance and an increase in dependency in the early years after diagnosis. Few differences were observed between the beginning and end of the study in the older age groups, suggesting a natural course of aging. The deterioration of ADLs and physical performance is mainly related to age (Romero et al., 2023) [20].

Contrary to our results, which indicate no significant relationship between functionality and pathological diagnosis, a systematic review of 43 studies with 19,246 patients suggests that one-third of adults with cancer need help with basic activities of daily living (BADLs) and half with instrumental activities of daily living (IADLs) [21].

Regarding gender, our results do not show significant differences in the SPPB battery between men and women. This finding suggests that both genders perform similarly in the physical performance tests evaluated by the SPPB, which could be relevant for functional rehabilitation. The absence of significant differences implies that rehabilitation programs can be designed uniformly for both genders, without the need for gender-specific adjustments. This facilitates the implementation of rehabilitation interventions and ensures that both men and women can equally benefit from the same therapeutic approaches, thereby optimizing resources and efforts in the field of functional rehabilitation. This coincides with the study by Gonsalves et al. (2023), although in this case, they found significant differences between patients with a cancer diagnosis of less than 5 years and more than 6 years, suggesting that the neurotoxic effects of treatments affect more recent diagnoses [22].

Physical performance, evaluated using the SPPB scale, may be associated with respiratory muscle weakness and loss of mobility, as noted in the study by Vaz Fragoso et al. (2023). Our study also found a direct relationship between dyspnea, evaluated using the MRC scale, and age, tumor stage, and number of treatment lines, indicating that dyspnea increases with these factors [23].

Although no relationship was found between gender and pathological diagnosis with dyspnea, the study by Vaz Fragoso et al. (2023) suggests that the risk of dyspnea increases with age and sedentary behavior, similar to our findings. Our study included a younger sample, with a mean age of 53.75 years, and found that lung cancer patients experienced dyspnea, significantly improving their functionality after chemotherapy (Hong et al., 2023) [24].

Dyspnea and shortness of breath cause anxiety and fear of movement (kinesiophobia), worsening physical performance and quality of life. Our study found a directly proportional relationship between age and quality of life in patients with dyspnea, confirming that advanced age carries a higher risk of dyspnea.

Studies such as Pilegaard et al. (2023) indicate that dyspnea negatively affects the ability to perform occupations, associating it with other symptoms such as pain and fatigue. In the advanced stages of the disease, these occupations become secondary, highlighting the importance of occupational therapy in clinical management [25].

Regarding quality of life, evaluated using the ECOG scale, our study indicates a direct relationship with age, number of treatment lines, and tumor stage. Patients with older age, advanced stage, and more treatment lines obtained a higher ECOG score, reflecting a worse functional state.

Other studies corroborate our findings, showing that patients with ECOG scores of 0 to 2 have better functional status and greater physical capacity, while those with scores of 2 to 4 show significant deterioration [26,27,28,29]. Additionally, a lower ECOG score and lower tumor stage are associated with higher survival (Romero et al., 2023) [20].

In conclusion, compared to other studies, such as Romero et al. (2023), which analyzed more than 600 cancer patients, four out of five reported a decrease in physical performance after diagnosis, with symptoms such as fatigue and pain closely related to reduced physical exercise [26]. 

It is important to clarify that correlation between variables does not imply causation; it simply indicates what is more likely in statistical terms. A correlation shows that there is a relationship between two variables, but not necessarily that one causes the other. Other underlying factors, such as improvements in overall health, may also contribute to these results. Therefore, it is crucial to interpret correlations with caution and consider the possibility of other variables influencing the observed relationship. We recommend conducting additional studies, especially those with experimental designs, to establish definitive causal relationships.

The study limitations include patient heterogeneity, the influence of socioeconomic status, and the subjectivity of dyspnea. Additionally, the inclusion of patients in advanced stages of the disease may affect the results. Future research should consider the specific type of treatment and the nutritional status of patients, including the presence of cachexia, as conditioning factors. Finally, we would like to highlight the need to improve future multicenter studies that address this secondary symptom from a more comprehensive perspective, as the characteristics of patients at a single center may bias the generalization of the results.

This study can be expanded to include additional variables that influence the quality of life of cancer patients and can be implemented in oncology units through occupational therapy to improve patient functionality and quality of life. Occupational therapy and rehabilitation play a crucial role in improving the quality of life for cancer patients experiencing dyspnea as a side effect. Occupational therapy helps patients adapt to their physical limitations and maintain independence in daily activities, while rehabilitation can include breathing exercises and fitness programs specifically designed to enhance lung function and reduce the sensation of breathlessness. Incorporating these interventions not only alleviates physical symptoms but also positively impacts the emotional and psychological wellbeing of patients, providing tools and strategies to better manage their condition and maintain a higher quality of life during cancer treatment.

## 5. Conclusions

Functionality and physical performance: Functionality and physical performance in cancer patients are inversely related to age, disease stage, and the number of treatment lines received. Specifically, older age, more advanced disease stage, and a higher number of treatments are associated with diminished functionality and poorer physical performance.

Dyspnea: Dyspnea in cancer patients is directly proportional to age, disease stage, and the number of treatment lines. Thus, increased age, more advanced disease stage, and a greater number of treatments elevate the risk of experiencing dyspnea.

Impact of dyspnea: Dyspnea significantly detracts from the functionality and physical performance of cancer patients, thereby negatively impacting their quality of life.

Gender and pathological diagnosis: The analysis reveals no significant relationship between functionality, physical performance, or dyspnea and the patient’s gender or pathological diagnosis.

Summary: Advanced age, more advanced tumor stage, and a higher number of treatment lines are strongly associated with reduced functionality, lower physical performance, and increased risk of dyspnea in cancer patients. Gender and specific cancer diagnoses do not influence these relationships.

## Figures and Tables

**Table 1 healthcare-12-01675-t001:** Descriptive statistics for the variables gender and age.

Features	Gender		
	Women (*n* = 85)	Men (*n* = 81)	Total (*n* = 166)
Gender	51.2%	48.8%	100%
Age			
Mean	67.02	66.62	66.82
Standard deviation (SD)	10,793	10,111	10,420
Minimum–maximum	50, 91	51, 85	50, 91

**Table 2 healthcare-12-01675-t002:** Descriptive statistics for the variables number of lines of treatment, anatomopathological diagnosis, and stage.

Features	Number of Patients (*n* = 166)	
	Frequency	Percentage
Number of treatment lines		
Two lines	8	4.8%
Three lines	114	68.7%
Four lines	39	23.5%
Five lines	3	1.8%
Six lines	2	1.2%
Pathological diagnosis		
Lung	100	60.2%
Digestive system	33	19.9%
Breast	25	15.1%
Central nervous system	3	1.8%
Prostate	2	1.2%
Other	3	1.8%
Stage		
Stage 2	2	1.2%
Stage 3	109	65.7%
Stage 4	55	33.1%

**Table 3 healthcare-12-01675-t003:** Descriptive statistics of the main variables under study.

Main Variables	Descriptive Statistics		
	Mean	Standard Deviation (SD)	Minimum–Maximum
Barthel Index	34.19	12,089	10–55
ECOG	2.83	0.666	2–4
mMRC	2.80	0.715	1–4
SPPB	5.66	2.041	1–11

**Table 4 healthcare-12-01675-t004:** Study of correlations of the variables under study.

Main Variables	Pearson Index (*p* = 0.05)		
	Age	Stage	Number of Treatment Lines
Barthel Index	−0.543	−0.373	−0.266
ECOG	0.561	0.276	0.220
mMRC	0.574	0.267	0.222
SPPB	−0.594	−0.282	−0.137

## Data Availability

The study data are available in the GREDOS repository of the University of Salamanca, within the scientific repository (https://gredos.usal.es/handle/10366/3823, accessed on 10 June 2024), and can be accessed by any researcher upon written request to the corresponding author of the manuscript.

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
