# Peer review of "Descriptive Study on the Relationship between Dyspnea, Physical Performance, and Functionality in Oncology Patients"

_healthcare, 2024, doi:10.3390/healthcare12161675_

Round 1

Reviewer 1 Report

Comments and Suggestions for Authors

GENERAL COMMENTS:  

The research article by Lucas-Ruano and colleagues reports on the impact of dyspnea on the functional status and physical performance of adult patients with cancer. This paper is important because functionality and physical activity are some of the most important modifiable risk factors that can improve the long-term outcomes and quality of life of cancer patients. In its current form, the study doesn’t really provide novel findings. I strongly suggest the authors consider more advanced statistical methods to strengthen their study. Outlined below are several concerns for the manuscript in its current form:

ABSTRACT:

Avoid using abbreviations in the abstract (ECOG, mMRC, SPPB)

INTRODUCTION:

-       Line 36: Cancer is a well-known disease. It’s unnecessary to define it. I would suggest directly starting the introduction by the epidemiology of cancer (line 40)

-       Line 43: It would be more sequential to cite the epidemiology of cancer in Spain (286,664) then specify which diagnoses are most common in the country.

-       Line 68: the sentence is a continuation of the previous idea regarding the causes of dyspnea. No need to start a new paragraph. 

-       Line 125: I really liked how the general and specific objectives were detailed. I would suggest however re-writing the specific objectives to have the predictor before the outcome in the sentence (example for objective 2: to evaluate the relationship between age (predictor) and dyspnea (outcome)).

METHODS:

-       Line 137: this sentence doesn’t make sense. It seems it was copied from an earlier version of a protocol. Please remove and start directly with the type of study (This is an observational cross-sectional study).

-       Line 140: which ethics committee? I would think it’s the ethics committee of the university. Please specify. 

-       Line 143: this seems to also have been copy pasted from the protocol. Please update to past tense (were selected, research was conducted).

-       Line 155: this is not an inclusion criteria to list. 

-       Line 172: past tense (was used)

-       Line 179: please add ats in line 56 to know what the abbreviation stands for.

-       Line 187, 189: also copy pasted from protocol. Please update. 

-       Line 190: who did the interviews? 

-       Line 194: what software was used? What security protection were in place to protect patient’s information? What will happen with the data once this study is published?

Statistical Analysis:

- BMI and Previous cardiovascular diseases are important confounders. Were they collected in this study?

Statistical tests used are very simple. The authors have a good number of participants and variables to conduct linear mixed methods to help elucidate the relationships between exposure and outcome (dyspnea).

RESULTS

-       Results very poorly written. Results section shouldn’t repeat what the tables and figures are showing but rather highlight the important information. Please re-write to highlight important relevant information in a paragraph style rather than individual sentences. 

-       Table 1: What is general? Total number of participants?

-       Table 3: what does y stand for?

Also please add footnote to remind reader what the abbreviations stand for. 

Comments on the Quality of English Language

Minor editing highlighted above. Mainly some information in the methods section were copy pasted from a protocol. I would highly recommend updating to the past tense. 

Author Response

Thank you for the insightful comments and suggestions. We have made the following revisions to the manuscript in response to your feedback:

ABSTRACT:

Avoid using abbreviations in the abstract (ECOG, mMRC, SPPB):

Response: Abbreviations are necessary in order to comply with the word limit.

INTRODUCTION:

Line 36: Cancer is a well-known disease. It’s unnecessary to define it. I would suggest directly starting the introduction by the epidemiology of cancer (line 40):

Response: We fully agree. We have summarized the sentence to the minimum possible.

Line 43: It would be more sequential to cite the epidemiology of cancer in Spain (286,664) then specify which diagnoses are most common in the country:

Response: We fully agree. We have modified the wording to make it more sequential and logical.

Line 68: the sentence is a continuation of the previous idea regarding the causes of dyspnea. No need to start a new paragraph:

Response: We completely agree. We have rewritten this section to be within the same paragraph.

Line 125: I really liked how the general and specific objectives were detailed. I would suggest however re-writing the specific objectives to have the predictor before the outcome in the sentence (example for objective 2: to evaluate the relationship between age (predictor) and dyspnea (outcome)):

Response: We fully agree. We have rewritten the study objectives accordingly.

METHODS:

Line 137: this sentence doesn’t make sense. It seems it was copied from an earlier version of a protocol. Please remove and start directly with the type of study (This is an observational cross-sectional study):

Response: We fully agree. We have deleted the confusing paragraph and modified the wording to start directly with the type of study.

Line 140: which ethics committee? I would think it’s the ethics committee of the university. Please specify:

Response: We fully agree. We have specified the ethics committee in the text.

Line 143: this seems to also have been copy-pasted from the protocol. Please update to past tense (were selected, research was conducted):

Response: We fully agree. We have updated the text to the past tense.

Line 155: this is not an inclusion criterion to list:

Response: We fully agree. We have modified the text of the manuscript.

Line 172: past tense (was used):

Response: We fully agree. We have modified the text to the past tense.

Line 179: please add ats in line 56 to know what the abbreviation stands for:

Response: We fully agree. We have added the explanation for the abbreviation.

Line 187, 189: also copy-pasted from protocol. Please update:

Response: We fully agree. We have modified the text of the manuscript.

Line 190: who did the interviews?

Response: We have included an explanation in the text regarding who conducted the interviews.

Line 194: what software was used? What security protection were in place to protect patient’s information? What will happen with the data once this study is published?

Response: We have included in the text an explanation of the software used for data storage and the security measures in place to protect patient information, as well as the plan for the data post-publication.

STATISTICAL ANALYSIS:

BMI and Previous cardiovascular diseases are important confounders. Were they collected in this study?

Response: Thank you very much for your consideration. While these variables were not collected in this study, we will take this into account in future research.

Statistical tests used are very simple. The authors have a good number of participants and variables to conduct linear mixed methods to help elucidate the relationships between exposure and outcome (dyspnea):

Response: Thank you very much for your consideration. We will consider more complex statistical methods in future studies.

RESULTS:

Results very poorly written. Results section shouldn’t repeat what the tables and figures are showing but rather highlight the important information. Please re-write to highlight important relevant information in a paragraph style rather than individual sentences:

Response: We fully agree. We have rewritten the results section to improve the understanding of the results and highlight the important information in a more cohesive paragraph style.

Table 1: What is general? Total number of participants?

Response: We have changed "general" to "total".

Table 3: what does y stand for?

Response: We have changed "y" to "-".

We believe these changes have significantly improved the manuscript and addressed your concerns. Thank you for your valuable feedback.

Best regards,

Reviewer 2 Report

Comments and Suggestions for Authors

1. The results seem to be presented in a cumbersome manner, could have done better.  

2. Need to explain more about the data in tables to help understand results better. What is normal, what is expected etc. 

3. Results were not statistically significant.  So it appeared more like an observational study.  

4. Introduction passage was good. 

Author Response

  1. The results seem to be presented in a cumbersome manner, could have done better.

Response: Thank you very much for your comments. We have improved the presentation of the results in the manuscript.

  1. Need to explain more about the data in tables to help understand results better. What is normal, what is expected etc.

Response: Thank you very much for your comments. We have improved the presentation of the tables to make them easier to understand.

  1. Results were not statistically significant. So it appeared more like an observational study.

Response: We fully agree, it is an observational study.

  1. Introduction passage was good.

Response: Thank you very much for your words and for your valuable time reviewing the manuscript, we are sure that after the changes it has improved its quality.

Reviewer 3 Report

Comments and Suggestions for Authors

The absctract should be modified and include all relevant aspects of the work; the objective of the work is not appreciated and the method section is especially deficient. 

Write the objective of the work in a single paragraph, avoiding the presence of ‘general’ and ‘specific’ objectives, which are typical of academic works, but not of scientific publications.

Lines one hundred and thirty-seven and one hundred and thirty-eight are superfluous. delete them. 

Because it uses convenience sampling in a cross-sectional study. Justify the answer appropriately. 

Have you made any calculations about the sample to be included?

How did you know how many people to collect? 

Was the study time determined in advance? 

What is the reason for such a long study period (more than three years)?

Eliminate the inclusion criteria, which allude to the voluntariness of the participants. This is a condition without equa-non. 

Strongly modify the exclusion criteria. The ones you use are not exclusion criteria. 

The wording of the results is confusing and the relevance of the results is not appreciated. 

Write the conclusions in an affirmative way about the findings that emerge from your study; avoid explicitly including aspects that do not emerge from the findings. 

Please avoid and eliminate grey evidence in your references. Please include indexed references on the definition of dyspnoea, or cancer; I cannot conceive of the use of grey evidence in such references.

Author Response

Thank you very much for your valuable comments and suggestions. We have made the following revisions to the manuscript in response to your feedback:

ABSTRACT:

The abstract should be modified and include all relevant aspects of the work; the objective of the work is not appreciated and the method section is especially deficient:

Response: Thank you very much for your comments. We have revised the abstract to include all relevant aspects of the work, ensuring the objective is clearly stated and the method section is adequately detailed.

OBJECTIVES:

Write the objective of the work in a single paragraph, avoiding the presence of ‘general’ and ‘specific’ objectives, which are typical of academic works, but not of scientific publications:

Response: Thank you very much for your comments. We fully agree and have rewritten the objectives in a single paragraph, eliminating sub-sections and improving the wording for clarity.

METHODS:

Lines 137 and 138 are superfluous. Delete them:

Response: We have removed these sentences as suggested.

Because it uses convenience sampling in a cross-sectional study. Justify the answer appropriately. Have you made any calculations about the sample to be included? How did you know how many people to collect? Was the study time determined in advance?:

Response: We have modified this section to provide a more robust justification for the sample size used. We have explained the sample size calculations and the predetermined study period.

What is the reason for such a long study period (more than three years)?:

Response: The justification for the extended recruitment period is due to two factors: first, the clinical characteristic of the dyspnea symptom, making it difficult to recruit such patients; second, the global COVID-19 pandemic, which hindered the recruitment process.

Eliminate the inclusion criteria, which allude to the voluntariness of the participants. This is a condition sine qua non:

Response: We have removed these criteria from the manuscript.

Strongly modify the exclusion criteria. The ones you use are not exclusion criteria:

Response: Thank you very much for your comments. We fully agree and have modified the exclusion criteria in the text according to your indications.

RESULTS:

The wording of the results is confusing and the relevance of the results is not appreciated:

Response: Thank you very much for your comments. We have rewritten the results section to make it clearer and highlight the relevance of the findings.

CONCLUSIONS:

Write the conclusions in an affirmative way about the findings that emerge from your study; avoid explicitly including aspects that do not emerge from the findings:

Response: Thank you very much for your comments. We have modified the wording of the conclusions to clearly state the findings from our study and avoid including aspects not directly derived from the results.

REFERENCES:

Please avoid and eliminate grey evidence in your references. Please include indexed references on the definition of dyspnea, or cancer; I cannot conceive of the use of grey evidence in such references:

Response: We have revised the bibliography to replace grey evidence with indexed references as suggested.

We believe these changes have significantly improved the manuscript and addressed your concerns. Thank you for your valuable feedback.

Best regards,

Reviewer 4 Report

Comments and Suggestions for Authors

Dear Authors,

Thank you for submitting your manuscript titled "Descriptive study on the relationship between dyspnea, physical performance, and functionality in oncology patients" for review. I appreciate the effort and thoughtfulness that went into conducting this important research. After a thorough review, I have identified several areas where the manuscript can be improved to enhance its clarity, rigor, and overall impact. Kindly consider addressing the following comments:

Sample Representation and Generalizability:

The study uses a convenience sample from a single hospital, which may limit the generalizability of the findings. Please discuss this limitation more explicitly in the manuscript and suggest ways future studies could address this issue, such as by using multi-center sampling.

Confounding Factors:

Potential confounding factors, such as comorbidities, socioeconomic status, and baseline functional status, are not extensively discussed. Acknowledge these factors and consider their impact on your findings. Including a multivariate analysis to control for these variables would strengthen the study.

Correlation vs. Causation:

While the study reports correlations between various factors and dyspnea, physical performance, and functionality, it is crucial to clarify that correlation does not imply causation. This distinction should be emphasized in both the methods and discussion sections.

Statistical Analysis and Interpretation:

Provide a more detailed explanation of the statistical methods used and their appropriateness for the data. Report effect sizes and discuss their clinical relevance to provide a better understanding of the practical implications of your findings.

Non-significant Findings:

The manuscript reports no significant relationship between gender or pathological diagnosis and the studied variables but does not discuss these findings in depth. Please provide a more thorough discussion of these non-significant results, exploring possible reasons and their implications for your hypotheses and future research.

Literature Review and Context:

The introduction could benefit from a more comprehensive review of the existing literature. Expand the literature review to include more studies related to dyspnea, physical performance, and functionality in oncology patients, highlighting the gaps your study aims to address.

Ethical Considerations:

Elaborate on how patient confidentiality was maintained and how any potential risks to participants were mitigated.

Discussion and Implications:

Expand the discussion to include a more detailed interpretation of the results, their implications for clinical practice, and specific recommendations for future research based on the study’s findings.

Study Limitations:

Provide a more comprehensive discussion of the study’s limitations, including potential selection and measurement biases, and how they might have impacted the results.

Thank you for considering these suggestions. I look forward to seeing the revised version.

Best regards,

Comments on the Quality of English Language

Consistency in Verb Tense:

The manuscript contains inconsistencies in verb tense usage, alternating between past and future tenses. To maintain clarity and professionalism, please ensure that the past tense is used consistently to describe completed actions, such as the study design, data collection, and analysis. The present tense should be used for general facts and interpretations, while the future tense can be reserved for recommendations and future directions.

Example corrections include:

"Patients were selected through a convenience sample..."

"Data were analyzed using the SPSS software..."

"The study showed that advanced age is associated with decreased functionality..."

.............................etc.

Author Response

Thank you very much for submitting your manuscript titled "Descriptive study on the relationship between dyspnea, physical performance, and functionality in oncology patients" for review. I appreciate the effort and thoughtfulness that went into conducting this important research. After a thorough review, I have identified several areas where the manuscript can be improved to enhance its clarity, rigor, and overall impact. Kindly consider addressing the following comments:

Sample Representation and Generalizability:

The study uses a convenience sample from a single hospital, which may limit the generalizability of the findings. Please discuss this limitation more explicitly in the manuscript and suggest ways future studies could address this issue, such as by using multi-center sampling:

Response: Thank you very much for your contributions. We are in complete agreement. We have included a reflection on this limitation in the manuscript and suggested multi-center sampling for future studies.

Confounding Factors:

Potential confounding factors, such as comorbidities, socioeconomic status, and baseline functional status, are not extensively discussed. Acknowledge these factors and consider their impact on your findings. Including a multivariate analysis to control for these variables would strengthen the study:

Response: Thank you very much for your contributions. We are in complete agreement. We have included a discussion on these potential confounding factors in the manuscript. Moreover, we will consider this as a fundamental section in future studies.

Correlation vs. Causation:

While the study reports correlations between various factors and dyspnea, physical performance, and functionality, it is crucial to clarify that correlation does not imply causation. This distinction should be emphasized in both the methods and discussion sections:

Response: We fully agree with this statement and have included an explanatory paragraph in both the methods and discussion sections of the manuscript to clarify this concept. Thank you for your consideration.

Statistical Analysis and Interpretation:

Provide a more detailed explanation of the statistical methods used and their appropriateness for the data. Report effect sizes and discuss their clinical relevance to provide a better understanding of the practical implications of your findings:

Response: Thank you for your consideration. We have redrafted the statistical methodology section to improve its explanation and included effect sizes to discuss their clinical relevance.

Non-significant Findings:

The manuscript reports no significant relationship between gender or pathological diagnosis and the studied variables but does not discuss these findings in depth. Please provide a more thorough discussion of these non-significant results, exploring possible reasons and their implications for your hypotheses and future research:

Response: Thank you for your consideration. We have expanded the discussion of these non-significant findings, exploring possible reasons and their implications for our hypotheses and future research.

Literature Review and Context:

The introduction could benefit from a more comprehensive review of the existing literature. Expand the literature review to include more studies related to dyspnea, physical performance, and functionality in oncology patients, highlighting the gaps your study aims to address:

Response: Thank you for your consideration. We have expanded the literature review to include more studies related to our research topic, highlighting the gaps our study aims to address.

Ethical Considerations:

Elaborate on how patient confidentiality was maintained and how any potential risks to participants were mitigated:

Response: We have added this information in the specific section "Informed Consent Statement" to elaborate on how patient confidentiality was maintained and potential risks were mitigated.

Discussion and Implications:

Expand the discussion to include a more detailed interpretation of the results, their implications for clinical practice, and specific recommendations for future research based on the study’s findings:

Response: Thank you for your consideration. We have expanded the discussion to provide a more detailed interpretation of the results, their implications for clinical practice, and specific recommendations for future research.

Study Limitations:

Provide a more comprehensive discussion of the study’s limitations, including potential selection and measurement biases, and how they might have impacted the results:

Response: Thank you for your consideration. We have included a more comprehensive discussion of the study’s limitations, including potential selection and measurement biases, and their possible impact on the results.

Thank you for considering these suggestions. I look forward to seeing the revised version.

Response: Thank you very much for your considerations. We have addressed each and every one of them. We hope that the new version will meet with your approval, and we greatly appreciate the time you have spent on it, as well as the improvement of the manuscript once we have made the changes.

Comments on the Quality of English Language:

Consistency in Verb Tense: The manuscript contains inconsistencies in verb tense usage, alternating between past and future tenses. To maintain clarity and professionalism, please ensure that the past tense is used consistently to describe completed actions, such as the study design, data collection, and analysis. The present tense should be used for general facts and interpretations, while the future tense can be reserved for recommendations and future directions:

Response: Thank you very much for your suggestions. We have conducted a thorough revision of the text to modify sentences with inconsistent verb tense usage and to ensure clarity and professionalism throughout the manuscript.

We believe these changes have significantly improved the manuscript and addressed your concerns. Thank you for your valuable feedback.

Best regards,

Round 2

Reviewer 1 Report

Comments and Suggestions for Authors

Thank you for considering my comments. I would suggest re-writing the 'analysis of variables' section in a more narrative manner. 

Comments on the Quality of English Language

N/A

Author Response

Thank you for considering my comments. I would suggest re-writing the 'analysis of variables' section in a more narrative manner.

Response: We greatly appreciate your valuable comments and suggestions, which have significantly contributed to improving the quality of our article. We have implemented the latest recommendations, making the necessary changes to the manuscript text. We sincerely appreciate the time and effort you have dedicated to reviewing our work, and we are confident that these improvements enhance the content and presentation of the study.

Reviewer 4 Report

Comments and Suggestions for Authors

Thank you so much for addressing the comments

Author Response

Thank you so much for addressing the comments

Response: We greatly appreciate the time and effort you have dedicated to reviewing our article. Your comments and observations are highly valuable to us, and we are very grateful for your detailed attention to the work. Your contribution is essential in improving the quality of the manuscript, and we sincerely thank you for your collaboration.
